# Parental Experiences of Melatonin Administration to Manage Sleep Disturbances in Autistic Children and Adolescent in the UK

**DOI:** 10.3390/healthcare11121780

**Published:** 2023-06-16

**Authors:** Jade Horsnell, Stephen Mangar, Dagmara Dimitriou, Elizabeth J. Halstead

**Affiliations:** 1Sleep Education and Research Laboratory, UCL Institute of Education, 25 Woburn Square, London WC1H 0AA, UK; jade.horsnell@nhs.net (J.H.);; 2Psychology and Human Development, UCL Institute of Education, London WC1H 0AA, UK; 3Department of Clinical Oncology, Imperial College Healthcare NHS Trust, Charing Cross Hospital, London W6 8RF, UK

**Keywords:** autism spectrum disorder, sleep disorder, child and adolescents, melatonin

## Abstract

Background: Autistic children and adolescents are 40–80% more likely to experience sleep disturbances than their neurotypical peers. In the United Kingdom, melatonin is licensed for short-term usage in adults at age 55 years and above; however, it is often prescribed to autistic children and adolescents to help manage their sleep. The current study sought to understand parental experiences and their motivation of using melatonin to manage sleep disturbances of their autistic children. Methods: The sample included 26 parents who took part in online focus groups answering questions regarding their experiences of using melatonin as a sleep treatment for their children diagnosed with autism between 4 and 18 years old. Results: Four main themes were identified: (i) parental perception of melatonin used as ‘a naturally produced hormone’; (ii) perceived benefits of using melatonin to improve their child’s sleep; (iii) administration of melatonin: dosage amount, timing and pulverising; and (iv) expectation and apprehension over melatonin use. Conclusion: Some parents reported success with the use of melatonin, and others reported the effects being limited or diminishing in time. Suggestions for healthcare professionals and families regarding melatonin usage in the UK are made with respect to setting clear guidelines for usage, whilst ensuring expectations are set and managed appropriately.

## 1. Introduction

Autism is a neurodevelopmental condition characterised by impairments in social cognition and behaviour affecting 1 in 54 children in the UK [1]. Young autistic individuals are more likely to experience a range of sleep disturbances including sleep onset delay, more frequent and prolonged nocturnal wakings and parasomnias than their typically developing peers [2]. Sleep disturbances have been associated with reduced social competence [3] and a general increase in autism scores [4], which in turn can impact their cognitive and behavioural functioning [5].

The treatment and management of sleep problems is crucial not only for autistic children but also for the well-being of their families [6]. Several non-pharmacological interventions have been utilised as an initial step in managing sleep disorders with a varying degree of success [7]. These include behavioural modification techniques such as establishing a consistent and positive bedtime routine, reducing screen time before bedtime and exercising during the day [8].

Guidelines from the National Institute of Clinical and Healthcare Excellence (NICE) in collaboration with the Social Care Institute (SCIE) (2013) suggested offering medication alongside the development of a sleep management plan using sleep diaries to help establish regular night-time sleep patterns [9]. These medications are required to be prescribed following a consultation with a paediatrician or psychiatrist, for short term use, and should be monitored regularly to ensure any benefits outweigh potential adverse effects.

Melatonin is a hormone secreted by the pineal gland and plays an important role in regulating the sleep/wake cycle [10]. Whilst the mechanism of sleep wake regulation in autistic children remains poorly understood, it has been suggested that deficiencies in melatonin synthesis may be an important factor [11]. In the UK, melatonin is licenced for short-term use in treating adults over the age of 55 years in the form of a prolonged release tablet—Circadin (Neurim Pharmaceuticals, Tel Aviv-Yafo, Israel). In addition, melatonin can also be utilised under specialist supervision for treating sleep difficulties in individuals with ASD, in circumstances where sleep hygiene measures have been exhausted and deemed insufficient [12].

Previous studies have tended to focus on immediate release formulations, syrups and crushed tablets; these studies have been largely retrospective with underpowered sample sizes, and thus insufficient, so far, to make any firm recommendations [13]. In particular, issues such as the long-term side effects of melatonin on behaviour and cognition in children and adolescents and the use of immediate release formulations remain unknown [14]. Furthermore, reports of positive outcomes may not be durable with the beneficial effects of melatonin diminishing after 3–12 months, though this can be enhanced if used in conjunction with light therapy [15].

Since 2018, a prolonged release mini-tablet paediatric formulation has been licensed for use, allowing primary care physicians to prescribe specifically for children who have insomnia associated with ASD where sleep hygiene measures are insufficient [16]. The long- and short-term side effects of the paediatric prolonged-release preparation on sleep health and behaviour have been investigated and data have been published from well-controlled clinical trials [17,18,19].

Whilst consensus guidelines of the British Association of Pharmacology [16] support the use of melatonin in combination with a behavioural intervention, more recently, NICE guidance acknowledges that there is still a lack of convincing evidence to suggest that administration of melatonin improves daytime behaviours and cognitive functioning of children and young adults [20]. This is despite reported high parental satisfaction [18,21].

This study sought to gain a greater understanding of parent’s attitudes and experiences of using melatonin for sleep disturbances of their autistic children’s sleep. In particular, the study explored parent and child interactions with healthcare professionals in terms of information provided during melatonin prescription and subsequent monitoring of usage.

## 2. Materials and Methods

### 2.1. Participants

A total of 26 parents (25 mothers and 1 father) of autistic children aged between 4 and 18 years old (10 females and 16 males) participated in the current study. Eligibility for the study required all children to have been prescribed melatonin for a minimum of three months to treat their sleep disorder/s by a paediatrician or psychiatrist. Children had all received a clinical diagnosis of autism spectrum disorder (ASD) and parents were living habitually with the child in the same household in the UK.

### 2.2. Procedure

Ethical approval was gained from the UCL IOE Research Ethics Committee (REC: 1227). Parents were recruited via adverts circulated on several different UK non-profit organisations and social media platforms. Potential participants contacted author JH, participants were screened as per the eligibility criteria, and informed consent was gained. Two asynchronous focus groups were conducted online using Piazza (www.piazza.com), each including 10–13 participants. Participation was anonymous during the focus groups as participants could enter Piazza without providing their name. Parents were asked to recall their child’s sleep patterns, specifically, the estimated time between going to bed and actually falling asleep (sleep onset). They were presented with 11 pre-determined questions within Piazza that asked about three key areas: (1) parent’s experiences of using melatonin with their child, (2) parent’s experiences of being prescribed melatonin for their child, and (3) guidance given to parents when using melatonin for their child (see Appendix A for full focus group questions). Participants were provided with a 14-day period to answer the questions and respond to comments from the moderators. The moderators commented on responses with prompts of requested clarity. The moderators were trained in focus group methodology by University College London (UCL). There was no required order for the questions to be answered and participants could return to their answers at any time. Parents were able to reply to other parent’s responses, allowing for discussion as would normally occur in face-to-face focus groups.

### 2.3. Data Analysis

To ensure rigour within results, Consolidated Criteria for Reporting Qualitative Research guidelines were implemented [22]. All written responses in each focus group were used verbatim for analysis. Since there was not enough information in previous literature to generate specific hypotheses, a grounded theory approach was selected to identify emergent coding categories. The constant comparative analysis was used to identify and code themes that emerged from the data. Two authors (J.H. and L.H.) independently read and coded the word document to identify concepts in the text. The authors reviewed and agreed upon their initial coding and categorisation, then organised codes into overarching themes. All themes/subthemes had a minimum of 10% or 3 participants. Final themes and subthemes were agreed upon by all authors.

## 3. Results

The demographic information on the study cohort with regards to age, ethnicity, social status and diagnostic details are summarised in Table 1.

All parents reported their child’s primary sleep disturbance was sleep onset delay and 15 parents (57.7%) reported their child had regular wakings during the night. Melatonin in tablet form was used by 21 families (80.7%) with 19 families using prolonged release and 2 families not knowing the preparation prescribed. Liquid melatonin was used by three families (11.5%) and two families used both tablets and liquid melatonin (7.7%). In addition, 16 (61.5%) of parents reported their child was also currently taking other medications for conditions such as anxiety, attention deficit hyperactivity disorder (ADHD) and asthma. The responses to the questions asked are summarised under four broad themes:

### 3.1. Theme 1: Understanding of Melatonin: “A Naturally Produced Hormone”

Many parents stated they found it acceptable using melatonin with their child as it is a naturally produced hormone—‘*such a simple thing to help, and as its produced naturally I didn’t think anything was wrong by giving it to him’*. All parents used the term *‘natural’* or *‘naturally occurring*’ to describe melatonin, and for this reason deemed it safe to use with their child. Many parents also stated they were encouraged to use melatonin by other families with autistic children, or healthcare professionals, e.g., ‘*we were encouraged to use it because it is usually naturally occurring in the body, prompting the ‘off switch*’.

### 3.2. Theme 2: Perceived Benefits of Using Melatonin to Improve Child’s Sleep

All parents stated they would recommend ‘to at least try’ melatonin to a family in a similar situation, if they had exhausted all other options. Most parents reported that melatonin use improved children’s sleep onset, specifically, melatonin was perceived to have a calming effect on their children, leading to reduced hyperactivity in the evenings. Parents reported that they were able to move the child’s bedtime earlier; some parents reported this was by 15 min, and some up to 2 h earlier. In addition, parents also reported settling down easier in their beds and better sleep onset, e.g., ‘*it has not completely resolved all sleeping issues but has significantly helped…particularly with time to fall asleep’*, *‘the main benefits were that [my child] would fall asleep noticeably quicker than before he had started taking it...*’. Although several parents reported that their child still has difficulties settling, e.g., ‘*Even now, with melatonin and aged 11, he cannot settle himself to sleep we have to sit with him*’. Improved sleep onset often alleviated strain placed upon parents and siblings, especially those who shared a room with their autistic sibling. Parents reported an increase in quality family time in the evenings and that they experienced less overall stress, as their own sleep was improved, e.g., ‘*overall, we as a family have a better quality of an evening/sleep for all of us and some downtime for me and my partner...*’.

Although most parents reported improvements in sleep onset whilst using melatonin, some parents also noted that their child still presented with early morning waking, e.g., ‘*Although he still wakes early, he falls to sleep within half an hour of taking it*’. Finally, parents reported that the positive effects of melatonin use diminished over time, and for some, melatonin stopped having any positive effect on sleep onset, e.g., ‘*The first night we (Mummy & Daddy) thought it was an absolute wonder drug... that quickly wore off! quite literally!’* and *‘It does seem like it is not having the effect it used to*’. Parents speculated reasons for this could be their child’s body becoming ‘use’ to melatonin and dosages needing to be increased with age, e.g., ‘*I am now finding it less effective than it was to begin with. I am unsure as to whether that is due to my son growing bigger or that he’s getting used to it*’. Parents described that the effects of melatonin plateaued or diminished, although no conclusive timeframe could be determined as to when effects diminished occurred, as the times varied between families (e.g., weeks vs. months). Identifying this concern to an HCP often resulted in increased dosage. Some parents reported no effects of melatonin for improved sleep, e.g., ‘*Our daughter had no side effects, but the melatonin didn’t really work,*’ and ‘*We were hopeful that it would work so we were disappointed that it didn’t*’.

### 3.3. Theme 3: Administration of Melatonin: Dosage Amount, Timing and Pulverising

#### 3.3.1. Dosage Amount

Parents actioned HCP’s advice by increasing the dosages given to their child. Parents often experimented with various dosages, until they found a quantity that worked for their child—this was sometimes referred to as ‘trial and error’. When first using melatonin, most parents started with a dosage of 2 mg and this was frequently increased, if required, to a dosage anywhere between 5 and 10 mg. Parents reported needing to increase the dosage over time to obtain the same results as when administration started, e.g., ‘*we started on a lower dose—maybe 2 tablets?—and gradually increased to 5 tablets at bedtime*’. Parents and children had concerns surrounding increasing the dosage as children were worried that it made them feel overly lethargic—‘*We have been told to up the dosage if needed, but my son has voiced his concerns about being too sleepy from a higher dose*’.

#### 3.3.2. Time of Administration

Some parents stated they were instructed by HCPs to give melatonin 30 min—1 h before their child’s bedtime—‘*we were told to give it to [my child] around 30 min—1 h before [their] desired bedtime*’. Some parents reported having no instructions for administration of melatonin when prescribed, e.g., ‘*we have never been given any instructions at all!*’.

#### 3.3.3. Pulverising Tablets: Sensory and Medical Issues

Parents of children who also had sensory processing difficulties or swallowing difficulties reported challenges around their child swallowing tablets. HCPs recommended parents crush the tablet and mix it with other foods such as yoghurt or juice—‘*with sensory issue’s [my son] wouldn’t even entertain the idea of the tablet crushed into powder with water in a syringe... so it’s become a real battle to get it into him*’. Alternatively, some parents found success in crushing the tablets, and some recognised this would mean the melatonin was no longer slow release: ‘*Although the instructions say they are not to be crushed, after discussion with their paediatrician... he agreed that this was ok and in fact I think is the only way I could get them to take it*’. and ‘This has zero effect, so we have been advised to crush the tablet which destroys the time release coating*’*. However, other HCPs had advised parents against crushing melatonin tablets: *‘Other than the chemist situation not liking us using it or crushing it*’.

### 3.4. Theme 4: Expectation and Apprehension over Melatonin Use

Many parents reported hoping initially that melatonin would solve their child’s sleep disturbances, specifically night waking and early morning waking, e.g., ‘*we thought it might help keep him in bed all night, but this wasn’t the case*’ and give their child the sleep pattern of a neurotypical child, e.g., ‘*when we were first prescribed it, I literally hugged our lead paediatrician with the idea it was going to ‘do what it says on the tin*’. Several parents reported disappointment after using melatonin for several months ‘*Sleep problems are so debilitating I’d still recommend others to try it but not to pin all their hopes on it as I did initially*’ and ‘*it wasn’t the magic wand we had been lead to believe it may be*’. Some parents were concerned their child may become dependent on melatonin to fall asleep. Overall, some HCPs were reported by parents to be supportive of melatonin use; however, some parents reported their HCP would disagree with the parent choice to request melatonin. This led to parents examining their own decision to use melatonin and question if it was the best option for their child, e.g., ‘*I do wish that some Drs would not be quite so judgemental when asking for more (Not all GP’s, some have been so supportive)’* and *‘I knew we were in despair when we started the process, I wish we’d been braver to ask earlier but everyone (HCPs) was so judgemental*’.

## 4. Discussion

This qualitative study highlights some important observations on the experiences and attitudes of the use of prescribed melatonin by parents for their children with ASD. There was high parental satisfaction with melatonin, especially with regards to a subjective improvement in sleep onset for their children. Nevertheless, parents encountered limitations and challenges with its use. Although melatonin generally seemed to help children settle in the evenings, this was at the expense of more night wakings and earlier morning waking. However, this study cannot provide a firm conclusion about whether this was the case and, instead, early mornings wakings could be the result of an improved overall quality of sleep including night time wakings.

A common theme reported was difficulty in administering the drug in tablet form due to sensory processing difficulties. Parents found advice from HCP’s conflicting and/or judgemental; however, the motivation for using melatonin was often driven by external factors such as positive experiences of other families.

Parents did not report concerns about using melatonin, for their children, as a drug with limited proven benefit and a lack of data concerning its safety and side-effect profile [23] because they viewed melatonin as a ‘natural product’. However, interestingly, none of the parents reported having an understanding about the role of melatonin and circadian rhythms, and parents lacked awareness of how melatonin works naturally within the body. This is probably not surprising given that, within the scientific and clinical community, its exact role and interaction with developmental age is yet to be fully elucidated. Thus far, research has identified that melatonin plays a significant role in other biological mechanisms and interacts with enzymes such as superoxide dismutase and catalase, as well as being closely linked to heightened immune-inflammatory activity [24].

Whilst parents reported a positive effect of melatonin on sleep onset, there was acknowledgement that this effect was short-lived, and commonly resulted in either dosage withdrawal or increase, in keeping with recommendations made by their HCP. This is consistent with previous research which reported that children who use melatonin often experience an increase in night wakings compared to those who do not use melatonin [25]. Based on the current and previous research, it is difficult to establish if these difficulties were present prior to melatonin use or if they developed after the onset of melatonin use. Melatonin is predominantly used for sleep onset insomnia and delayed sleep–wake phase problems [24], both of which are highly prevalent, but only form part of the range of sleep disorders in autistic populations [26]. It is thus possible that the presence of a diverse range of sleep disturbances may reduce the potential benefits of melatonin treatment.

It would be helpful to characterise sleep disturbances, either subjectively with sleep diaries or more objectively with actigraphy, in an attempt to demonstrate insomnia or a delayed phase sleep syndrome, as well as to exclude other circadian sleep disorders before recommending use of melatonin. Due to the limited evidence of long-term melatonin treatment, little is known about the possible side effects of increased dosage over time. However, preliminary research has found that continued increase in dosage for children can lead to potential drug dependency requiring long-term treatment [23]. It was noted that parents were increasing the dose of the prolonged-release preparation as they were not seeing an instant response. This might be related to the fact that the prolonged-release tablet was not administered at least an hour before the habitual bedtime of the child. Therefore, regular monitoring of melatonin treatment by HCPs is essential. It seems reasonable that the use of prescribed melatonin alongside longer-term behavioural management strategies may be beneficial, and should be used alongside, and not as a replacement for, these strategies. For example, research has shown that interventions such as physical activity help improve sleep by increasing melatonin levels, as well as practicing and implementing optimal sleep hygiene [7]. Parents mentioned using an array of behavioural techniques to improve their child’s sleep, such as weighted blankets and white noise. As these techniques were used in conjunction with melatonin, it is challenging to distinguish whether the positive outcomes noted by parents were caused by melatonin usage alone.

A limitation of this study is that the sample size was achieved via opportunistic sampling; therefore, data saturation was not reached. However, this study provides insight into the variability of the use of prescribed melatonin, showing a lack of consistency of guidance by HCPs. Some parents encountered difficulties administering melatonin to their children, particularly if their child had sensory processing difficulties which are common in this population. Many parents found it challenging for their child to swallow tablets due to the feeling of the tablet in their mouths or the associated taste. Many HCPs suggested crushing the tablet and mixing it with a drink or yoghurt, although this is against the pharmacological instruction for the drug’s use. In addition, research has found that pulverising tablets significantly changes the drug release profile as it becomes immediate-release and is no longer consistent with the summary of product characteristics [27]. There is limited evidence to demonstrate that immediate-release melatonin is safe to use in child and adolescent populations, which therefore raises treatment and safety concerns [28]. It was also not clear from this study if families changed the administration time to account for pulverising the tablet, although one family reported their HCP had recommended adjusting administration time to 30 min before sleep. There was a lack of appreciation that melatonin is available in a liquid preparation and, in part, this may be restrictive because of cost implications.

Data gathered from the focus groups suggest that inconsistent advice was also provided on the timing of the drug administration, which is a key consideration in terms of its efficacy. Guidance provided to parents was seemingly not clear in that melatonin should be administered 30–60 min prior to the child’s habitual bedtime, and at times was inconsistent, with some HCPs recommending breaks from melatonin on the weekends or during school holidays, whereas others recommended continued use. This is despite studies suggesting that benefit of the paediatric dose maybe enhanced if given 1–2 h before the desired bedtime [19,29]. Neither of these recommendations were consistently adhered to in the current study. It would be beneficial to not only provide clear guidelines to families on how to use melatonin and manage expectations, but also to increase provider education on melatonin use in families of children with ASD. This may reduce the ‘trial and error’ type usage when parents attempt to ascertain how best to use melatonin. Furthermore, the duration of melatonin use varied widely, ranging from weeks to years. It would be of interest to conduct a follow-up study to distinguish if experiences change over time, specifically, to investigate effects of melatonin diminishing or plateauing. It was noted from this study that some of the children had comorbid ADHD, and this is consistent with the experiences from other countries where high rates of melatonin prescription were noted in children and adolescents with ADHD and ASD [24,30]. With these observations in mind, the authors have suggested a checklist that may be helpful for HCPs and families for prescribing and administering melatonin; see Figure 1.

## Figures and Tables

**Figure 1 healthcare-11-01780-f001:**
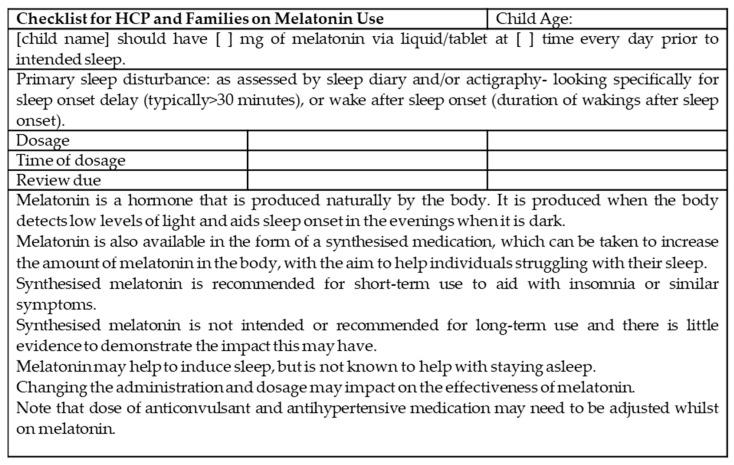
A suggested checklist for HCPs and families for melatonin use.

**Table 1 healthcare-11-01780-t001:** Participant demographic data.

Variable	Mean	Range	SD	*n*	Percent
Parental Age	44 years	32–57 years	6.8	-	-
Child Age	11 years	4–18 years	3.6	-	-
Time since ASD Diagnosis	4.2 years	1–15 years	3.2	-	-
Length of Melatonin Use	2.5 years	0.3–8 years	2.0	-	-
Melatonin Dosage	3.5 mg	2–10 mg	2.1	-	-
Relationship Status					
*Married*				23	88.5%
*Separated*	-	-	-	2	7.7%
*Divorced*				1	3.8%
Ethnicity					
*White British*				23	88.5%
*White European*				1	3.8%
*Asian*	-	-	-	1	3.8%
*Mixed race*				1	3.8%

## Data Availability

Data are unavailable due to privacy or ethical restrictions.

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
