# Peer review of "Parental Experiences of Melatonin Administration to Manage Sleep Disturbances in Autistic Children and Adolescent in the UK"

_healthcare, 2023, doi:10.3390/healthcare11121780_

Round 1

Reviewer 1 Report

General comments: Thank you so much for giving me the opportunity to review this well written and structured manuscript. The authors have addressed an important issue in the ASD population, which is sleep disturbances and its management with melatonin through exploring parental experiences. The authors highlight the importance of providing clear guidelines to families regarding melatonin administration as well as a suggested checklist that can be used by health care professionals that ensures all the information about melatonin use is provided for families in order to reach the expected outcomes from its use.

However, I had some suggestions that would help in improving the manuscript.

Abstract: In line 13 “usage in adults over the age of 55 and over”, I suggest saying usage in adults at age of 55 and above.

Introduction:

In line 35, I suggest adding a reference here for the impact of sleep disturbances on cognitive and behavioral functions.

In line 55, there is mention that melatonin has been used in patients with learning disabilities and behavioral problems, I was interested to know about its use in ASD children. It would be interesting to know if it is commonly used in this population, if this data is available to the authors.

Materials and methods

The authors did not provide reasoning for the sample size included in the study. I would suggest providing an explanation for the number of participants included. One of the determining factors is data saturation, when the authors sense that they are not getting any new data from focus groups. However, the current study included only 2 focus groups each with 13 participants.

In line 87, the authors mention that participation was anonymous. It would be helpful to provide some clarification here, did the authors mean that participants were given pseudo names for example.  The authors clearly describe the methods of recruiting participants, but they didn’t provide information about how they reached out to participants as who reached out to them, how did they consent to participate in the study.

In line 92, I would suggest naming supplementary material as focus group questions because the material provided is questions not a schedule.

There is a need to provide more clarification about the procedures implemented in the study. The authors mention that they gave the parents the questions, how did they send the questions to the parents, was it through email or mail. Also, how did the authors receive the responses?

It was mentioned also that parents were allowed to discuss with each other their responses. How was this discussion moderated, did the moderator go over the questions again and allow parents to discuss their points of view together? This section needs more elaboration for the steps that have been done to be more clear to the reader. Also, what are the qualifications of the moderator? How long did each focus group last?  I was also wondering, given the large number of participants in each group, how the discussion between parents go and did each one of them contribute to the discussion.

Results:

In line 154: “Parents described that when the effects of melatonin plateaued or diminished, although a specific timeframe of this effect varied between families”. It wasn’t clear what the authors meant by this sentence.

Discussion

In line 283, “It 282 would be beneficial to provide clear guidelines to families on melatonin use of expectation of benefits”. Please revise this sentence to clarify the meaning.

I suggest adding a section at the end of the manuscript discussing the limitations of the current study as well suggestions for future research.

Author Response

Dear Reviewer, 

On behalf of the authors I would like to thank you for your insightful comments. Please find our latest manuscript and response letter. With Kind regards, Professor Dagmara Dimitriou

Reviewer 2 Report

It was a pleasure to read this study about melatonin treatment in autistic children.

The introduction and discussion were verry well written.

For the result part, we would expect some numbers. The results are descriptive and at the end, we don't know how much families are concerned as a lot of sentences starts as "some parents ...". It would be usefull to have results and have more objective criterias that allows to objectively evaluate each theme. Tables would be appreciated. 

The study subject is interesting but for me the result are unclear.

Author Response

(The authors gave the same response as above.)

Reviewer 3 Report

The authors have provided a well-written summary of a qualitative analysis of parental perceptions of melatonin administration for sleep problems in autistic youth. The qualitative methods are clearly detailed. The findings are interesting and the authors have included example quotations, which is appreciated in qualitative analyses. The difficulty administering the medication (i.e., need to crush and combine with food) was an interesting point that is especially relevant to autistic children, and not as often highlighted. The following are very minor points to improve clarity.

At line 55, the authors state that melatonin can be utilized for treating insomnia in patients with learning disabilities? Do they mean “developmental disabilities” here? When I read “learning disabilities” I think of dyslexia and dysgraphia, etc., rather than pervasive developmental disorders, like autism spectrum disorder.

In the references, can the authors update the link to the NICE guidelines? The organization likely updated their website, as the link directs to a “We can’t find this page” message.

In the Discussion, the following sentence is slightly misphrased: “Melatonin is predominantly used for circadian rhythm disorders such as sleep onset insomnia and delayed sleep onset…” For an easy fix, I suggest removing “…circadian rhythm disorders such as…” from this sentence. Sleep onset insomnia is not classed as a circadian rhythm disorder, though certainly some people do experience both sleep onset insomnia and delayed sleep-wake phase problems. Also, it may be helpful if the authors switch “delayed sleep onset” to “delayed sleep phase problems” or “delayed sleep-wake phase problems” if I’m understanding them correctly, as “delayed sleep onset” might sound like a different way of saying sleep onset insomnia to some.

Minor point: Line 238-239 could use rephrasing to be less speculative, as it counters the prior sentence. The prior sentence claims that sleep onset insomnia and delayed sleep onset are only a portion of the range of sleep problems in autistic individuals. However, the next sentence states “It is thus possible that a range of sleep disturbances are present in children that are not fully characterized therefore reducing the potential benefits of melatonin treatment.” It would help if the authors clarified whether they were referring to their own sample or the general literature. If referring to the general literature then these two sentences leave me wondering whether or not the full range of sleep problems in children with autism has been characterized/reported on. For example, a revised sentence might state: “Thus, these diverse sleep problems may reduce the potential benefits of melatonin treatment.”

The authors have provided a helpful checklist to educate families about melatonin. It may be helpful for the authors to discuss the clinical/educational implications of their study more overtly. For example, findings highlight the need for not only parent education, but provider education too.

Author Response

Dear Reviewer,

On behalf of the authors I would like to thank you for your time and your insightful comments. Please find our revision and response letter.

With Kind Regards,

Professor Dagmara Dimitriou

Reviewer 4 Report

This qualitative study highlights some important observations on the experiences and attitudes of the use prescribed melatonin by parents for their children with ASD. There are several points that call for the authors’ attention:

11.        While it appears that when the study was commenced there was no approved melatonin product for children with ASD, this has since changed  substantially with the approval of the pediatric preparation (minitablet) in 2018 providing the necessary efficacy and safety data in this population.  In particular, issues such as the short and long-term side effects of this pediatric prolonged release melatonin preparation on sleep, health, and behavior have in fact been investigated in well controlled clinical trials in children and adolescents with ASD and the data have been published.

22.       There is scarce evidence on long term safety of immediate release formulations, syrups and crushed tablets though. Some of these aspects are in fact covered by the cited literature and should be clarified in the paper.

33.       Another limitation is the missing information on the association between some of the observations and the type of melatonin used (immediate release, prolonged release, crushed tablets or syrup), as the preparation makes a difference in terms of phase advance, sleep maintenance and even desensitization (presumably in slow metabolizers).  If the authors have the data it should be included in the results and discussion sections.

These are major limitations in the paper and should be included in the introduction and discussion sections and overall conclusions of the study to assist in the clinical decision making with respect to choice of treatment for the sleep problem in the child and adolescent with ASD.   

Author Response

(The authors gave the same response as above.)

Round 2

Reviewer 2 Report

Thank you for your answer.

Author Response

Dear Reviewer please note that we have already responded to all your comments which we believe have improved out paper.

With Kind Regards, 

Professor Dagmara Dimitriou 

Reviewer 4 Report

This qualitative study highlights some important observations on the experiences and attitudes of the use prescribed melatonin by parents for their children with ASD. There are several points that call for the authors’ attention:

1.        While it appears that when the study was commenced there was no approved melatonin product for children with ASD, this has since changed  substantially with the approval of the pediatric preparation (minitablet) in 2018 providing the necessary efficacy and safety data in this population.  In particular, issues such as the short and long-term side effects of this prolonged melatonin preparation in the form of minitablets on sleep, health, and behaviour have in fact been studied in well controlled clinical trials in children and adolescents with ASD and the data have been published.

2.       There is scarce evidence on long term safety of immediate release formulations, syrups and crushed tablets though. Some of these aspects are in fact covered by the cited literature.

3.       Another limitation is the missing information on the association between some of the observations and the type of melatonin used (immediate release, prolonged release, crushed tablets or syrup), as the preparation makes a difference in terms of phase advance, sleep maintenance and even desensitization (presumably in slow metabolizers).  If the authors have the data it should be included in the results and discussion sections.

These are major limitations in the paper and should be included in the introduction and discussion sections and overall conclusions of the study to assist in the clinical decision making with respect to choice of treatment for the sleep problem in the child and adolescent with ASD.  

Author Response

(The authors gave the same response as above.)

Round 3

Reviewer 4 Report

For some reason I did not see the revised version, I still have major concerns and I am afraid these may confuse the reader.

1. The paper completely lacks quantitative information. It is quite difficult to draw conclusions if the reader cannot tell major from minor events. Most of my comments below are derived from the lack of firm reporting. As the authors seem to possess these data they should present them.
2. The NICE document was published before the pediatric form of prolonged release melatonin was introduced, and before the results of the relevant studies were published (Schroder et al 2019 for example). Therefore the sentence on NICE (Lines 73-77) belongs to the information provided in lines  59-65  before they discuss the pediatric form. The information on the daytime effects of the pediatric PRM form do not belong to the NICE paper.  As written now, it may mislead the reader to believe that  NICE has seen and disregarded the data in the later publications.
2. The loss of efficacy with time is reportedly seen  with immediate release melatonin rather than the prolonged release formulation. Most cases of loss of effectiveness (lines 173-184) might therefore be connected to crushing of the adult size PRM.  There is no information to check whether this was indeed the case.
3. The authors make a statement   (lines 243-244)   that " A common theme reported was difficulty in administering the drug in tablet form due to sensory processing difficulties and this is despite the smaller size (3 mm) of the 244 peadiatric modified tablets." where is the evidence? it seems that most children were using the adult size tablet rather than th eminitablet. It is anyway quite challenging to crush the minitablet.
4. There is much confusion about the dosing (lines 277-280). The authors have cited the Schroder Expert Opinion paper (18) but have not brought its contents into perspective. The paper presents the dose titration as the main strategy to be used, the parents were actually doing what is right and recommended.
5. The timing of administration (lines 314-318) as parents did is the right one and not as stated in the paper (30 minutes, not 1-2 hours, again as clearly stated in the Schroder papers).  
6. A minor comment- Circadin is allowed for 55 and older (as corrected in the Abstract) but is still not consistently corrected in the paper (e.g. line 53).

hope my comments will be considered by the authors.

Author Response

Dear Reviewer,

Please find our modified version of the paper. We hope that this is now in line with some of your comments. Thank you for your time on this paper.

With Kind Regards,

Professor Dagmara Dimitriou
